# Selective Labeling: How to Radically Lower Data-Labeling Costs for Document Extraction Models

**Yichao Zhou, James Bradley Wendt, Navneet Potti, Jing Xie, Sandeep Tata**
Google Research
Mountain View, USA
{yichaojoey, jwendt, navsan, lucyxie, tata}@google.com

## Abstract

Building automatic extraction models for visually rich documents like invoices, receipts, bills, tax forms, etc. has received significant attention lately. A key bottleneck in developing extraction models for new document types is the cost of acquiring the several thousand high-quality labeled documents that are needed to train a model with acceptable accuracy. In this paper, we propose *selective labeling* as a solution to this problem. The key insight is to simplify the labeling task to provide "yes/no" labels for candidate extractions predicted by a model trained on partially labeled documents. We combine this with a custom active learning strategy to find the predictions that the model is most uncertain about. We show through experiments on document types drawn from 3 different domains that selective labeling can reduce the cost of acquiring labeled data by $10\times$ with a negligible loss in accuracy.

## 1 Introduction

Visually rich documents such as invoices, receipts, paystubs, insurance statements, tax forms, etc. are pervasive in business workflows. The tedious and error-prone nature of these workflows has led to much recent research into machine learning methods for automatically extracting structured information from such documents (Lee et al., 2022; Garncarek et al., 2021; Xu et al., 2021; Tata et al., 2021; Wu et al., 2018; Sarkhel and Nandi, 2019). Given a target document type with an associated set of fields of interest, as well as a set of human-annotated training documents, these systems learn to automatically extract the values for these fields from documents with unseen layouts.

A critical hurdle in the development of high-quality extraction systems is the large cost of acquiring and annotating training documents belonging to the target types. The human annotators often require training not only on the use of the annotation tools but also on the definitions and semantics

of the target document type. The annotation task can be tedious and cognitively taxing, requiring the annotator to identify and draw bounding boxes around dozens of target fields in each document.

This data efficiency requirement has not gone unnoticed in the research literature on this topic. However, even with model pre-training (Xu et al., 2020, 2021), transfer learning from an out-of-domain labeled corpus (Torrey and Shavlik, 2010; Nguyen et al., 2019), and data-programming (Ratner et al., 2017; Zhang et al., 2022), empirical evidence suggests that performing well on a new target document type like procurement, banking, insurance, mortgage, etc. still requires thousands of annotated documents, amounting to hundreds of hours of human labor (Zhang, 2021).

The cost of acquiring high quality labeled data for hundreds of document types is prohibitively expensive and is currently a key bottleneck. We could apply active learning strategies to select a few but informative documents for human review (Settles, 2009), however the cost-reducing effect of this approach is limited, as it requires annotating the span in every selected document for every field. Many of these annotations are repetitive, and often not very informative if a model can already extract those fields easily. In fact, our initial experiments with a document-level active learning approach yielded modest results that cut down the number of documents required to get to the same level of quality as random selection by approximately 20%. In this paper, we propose a technique called *selective labeling* that reduces this cost by $10\times$. The key insight is to combine two ideas: First, we redefine and simplify the task performed by the human annotators – rather than labeling every target field in every document by drawing bounding boxes around their values, we ask them to simply verify whether a proposed bounding box is correct. This binary "yes/no" annotation task is faster and imposes a lighter cognitive burden on the an-

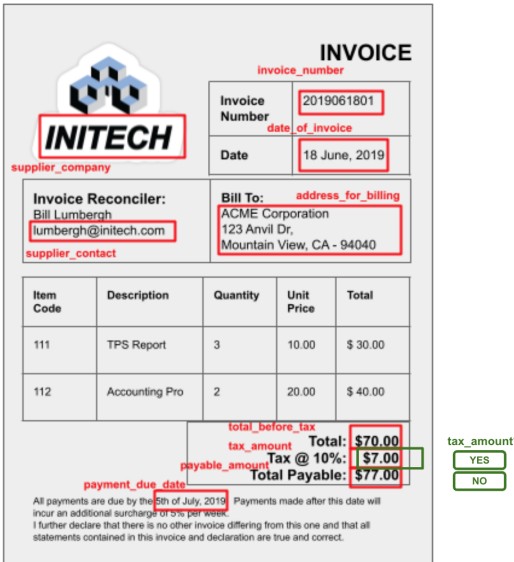

Figure 1: A classic annotation task: even labeling 9 fields in this toy invoice imposes a heavy cognitive burden on the annotator, while real-world documents are significantly more complicated (Red). A "yes/no" annotation task: presenting a proposed span and asking the annotator to accept or reject the label is simpler, quicker, and less prone to errors (Green).

notator (Boim et al., 2012; Blog, 2020; Ganchev et al., 2007; Skeppstedt et al., 2017). Second, we adapt existing active learning strategies to select the examples (i.e., candidate extraction spans) that the model is most uncertain in each round to annotate. In other words, we consider active-learning at the (document, field)-pair granularity rather than at the document level granularity and choosing a modeling approach that can easily deal with the complexity resulting from the partially labeled documents this approach produces.

We interleave rounds of such human annotation with training a model that is capable of consuming partially labeled documents. In combination, our proposed approach dramatically improves the efficiency of the annotation workflow for this extraction task. In fact, through experiments on document types drawn from multiple domains, we show that selective labeling allows us to build models with $10\times$ lower annotation cost while achieving nearly the same accuracy as a model trained on several thousand labeled documents. Note that our goal in this paper is *not* to advance the state-of-the-art in active learning, *nor* to propose a more data-efficient model for extraction from layout-heavy documents. Our main contribution is that we demonstrate that a novel combination of an existing active-learning strategy with an existing extraction model can be used to *dramatically cut down the primary bottleneck* in developing extraction models for visually rich documents.

## 2 Background

We first describe how a typical annotation task is set up to acquire labeled documents. We point out two major deficiencies with this approach before outlining an alternative that takes advantage of the characteristics of this domain. We then describe the assumptions underlying our approach.

### 2.1 Annotation Workflow

#### 2.1.1 Classic Annotation Workflow

Given a document type for which we want to learn an extraction model, we begin by listing out the fields that we want to extract, along with human-readable descriptions, viz., "labeling instructions". We provide these instructions to human annotators and present them with various document images to label. The classic annotation task is to draw a bounding box around each instance of any of the target fields and label it with the corresponding field name (Figure 1 Red Remarks). Typical document types like paystubs have dozens of fields, and each document may contain multiple pages.

The high cognitive burden of the classic annotation workflow leads to two major drawbacks. First, it makes training data collection extremely expensive. In one annotation task for paystub-like documents with 25 target fields, the average time to label each document was about 6 minutes. Scaling this to hundreds of document types with thousands of documents each would be prohibitively expensive. Second, the resulting annotation quality is often quite poor. We have observed systematic errors such as missing labels for fields that occur infrequently in the documents or for instances that are in the bottom third of the page. To obtain acceptable training and test data quality, each document must be labeled multiple times, further exacerbating the annotation cost issue.

#### 2.1.2 Proposed Annotation Workflow

We propose the following alternative to the classic annotation workflow:

1. We speed up labeling throughput by simplifying the task: rather than drawing bounding boxes, we ask annotators to accept or reject a candidate extraction. Figure 1 (Green

Remarks) illustrates how much easier this "yes/no" task is compared to the classic one.

2. We further cut down annotation cost by only labeling a subset of documents and only a subset of fields in each document.

3. We use a model trained on partially labeled documents to propose the candidate extraction spans for labeling. This allows us to interleave model training and labeling so that the model keeps improving as more labels are collected.

4. We use a customized active learning strategy to identify the most useful labels to collect, viz., the candidate extraction spans about which the model is most uncertain. In successive labeling rounds, we focus our labeling budget on the fields that the model has not yet learned to extract well, such as the more infrequent ones.

In Section 5, we show empirical evidence that this improved workflow allows us to get to nearly the same quality as a model trained on 10k docs by spending an *order-of-magnitude less* on data-labeling. Note that naively switching the labeling task to the "yes/no" approach does not cut down the labeling cost – if we were to highlight every span that might potentially be an amount and present an "Is this the tax_amount?" question, with the dozens of numbers that are typically present in an invoice, this workflow will be *much more* expensive than the classic one. A key insight we contribute is that a model trained on a modest amount of data can be used to determine a highly effective subset of "yes/no" questions to ask.

## 2.2 Assumptions

We make the following four assumptions about the problem setting:

1. We assume access to a pool of unlabeled documents. This is a natural assumption in any work on managing cost of acquiring labeled training data.

2. We assume the extraction model can be trained on partially labeled documents.

3. We assume the model can generate candidate spans for each field and a measure of uncertainty – this is used to decide the set of "yes/no" questions to present to the annotator.

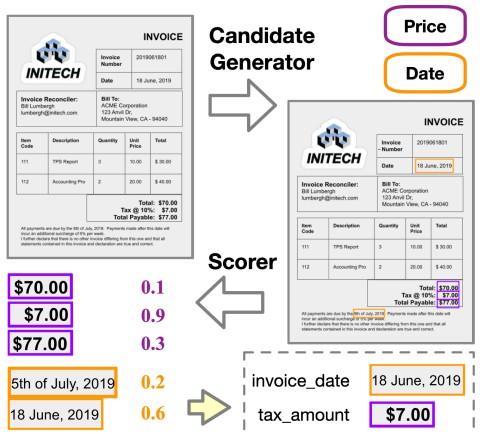

Figure 2: Architecture of the candidate generator and scorer of our document extraction model. The scoring is done using a neural network model trained as a binary classifier.

4. The analysis in this paper uses empirical measurements for labeling tasks on documents with roughly 25 fields to model the costs of the traditional approach (6 minutes per document, details in Appendix) and the proposed approach (10 seconds per "yes/no" question (Blog, 2020)). For more complex documents the difference in the two costs may be significantly higher.

Throughout this work, we use an extraction system similar to the architecture described in (Majumder et al., 2020). As shown in Figure 2, this architecture consists of two stages: candidate generation and candidate classification. In the first stage, we generate candidates for each field according to the type associated with that field. For example, the candidates generated for the *date of invoice* field would be the set of all dates in that invoice. The candidate generators for field types like dates, prices, numbers, addresses, etc. are built using off-the-shelf, domain agnostic, high-recall text annotation libraries. The recall of candidate generation varies across fields, e.g. high in dates and prices while relatively low in addresses and names. Having a candidate generator with low recall indeed limits the recall of the final extractions for that field. In the second stage, we score each candidate's likelihood of being the correct extraction span for the document and field it belongs to. This scoring is done using a neural network model trained as a binary classifier. The highest-scoring candidate for a given document and field is predicted as the extraction output for the document and field if it exceeds a certain field-specific threshold.

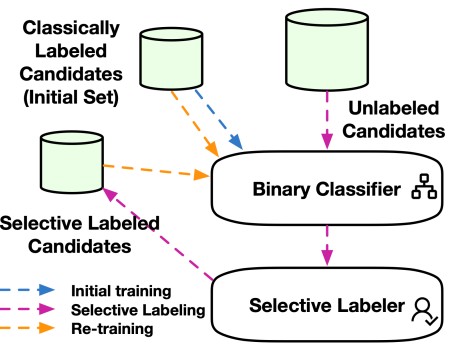

**Classically Labeled Candidates (Initial Set)**

**Unlabeled Candidates**

**Binary Classifier**

**Selective Labeled Candidates**

**Selective Labeler**

- - → **Initial training**
- - → **Selective Labeling**
- - → **Re-training**

Figure 3: The model training pipeline starts by inital training (blue) the binary classifier using the small classically labeled dataset. We then selectively label (purple) a fixed number of candidates according to the budget, which are then used to re-train (orange) the model together with the initial dataset.

The ability to train on partially labeled documents is trivially true for this modeling approach since it employs a binary classifier trained on the labeled candidates. This should be relatively straightforward for sequence labeling approaches, such as (Xu et al., 2021), as well. Identifying a potential span in the document to present as a "yes/no" question to an annotator is an exercise in ranking the candidates for each field. We expect that sequence labeling approaches can be adapted to satisfy this requirement, e.g., by using beam search to decode the top few sequence labels. However, this is likely more complex than the aforementioned approach, and we leave this as an exercise for future work.

## 3 Selective Labeling Methodology

We first provide an overview of the selective labeling framework before describing various uncertainty measures and ways to deal with the unique characteristics of this setting, such as varying difficulty for different fields.

### 3.1 Overview

Figure 3 provides a visual overview of our selective labeling workflow. We assume a scenario in which a corpus of several thousand unlabeled documents $U^d$ belonging to the target document type is available and we can request annotations from a labeler for every unlabeled document $d_i \in U^d$ which consists of a set of candidates $\{c_0^{d_i}, c_1^{d_i}, c_2^{d_i}, ..., c_n^{d_i}\}$. We begin by fully labeling a small randomly sampled subset of documents $S^d \subseteq U^d$, say 50-250 documents, using the classic annotation workflow. We learn an initial document extraction model $f(x|S^c)$, where $S^c$ represents the candidate set

contained in $S^d$ and we mark all the remaining unlabeled candidates in $U^d \backslash S^d$ as $U^c$. Our labeling workflow proceeds in rounds. In each round $j$, the model is used to select candidates $S_j^c$ from $U^c$ and have them reviewed by human annotators. The annotators answer a "yes/no" question either accepting or rejecting this proposed label. As a result, $S^c = S^c \cup S_j^c$ and $U^c = U^c \backslash S_j^c$, meaning the newly labeled examples are merged into the training set and removed from the unlabeled set. The model is retrained on $S^c$ in each round and we repeat this iterative labeling-and-training procedure until we exhaust our annotation budget or reach our target F1 score.

### 3.2 Measuring Uncertainty

We select the candidates that the model is *most uncertain* about. In this work, we explored two metrics to quantify a model's prediction uncertainty.

**Score distance.** This method assigns a metric to each candidate based on the distance that the score is from some threshold (Li and Sethi, 2006). More formally, the uncertainty is $1 - |score - threshold|$. For example, if the threshold is 0.5, this suggests that the model is most uncertain of its predictions of scores close to 0.5, in either direction. This approach can also be interpreted as an entropy-based uncertainty method, where we find an optimal candidate $x^*$ so that $x^* = \arg\max_x - \Sigma_i P(y_i|x)\log(P(y_i|x))$. In our binary classification setting, $y_i = \{0, 1\}$ and candidates with scores closer to 0.5 results in larger entropy.

**Score variance.** This method performs inference on a candidate multiple times with the dropout layer enabled and assigns the uncertainty metric as the variance of the scores (Gal and Ghahramani, 2016; Kirsch et al., 2019; Ostapuk et al., 2019). An alternative method trains multiple models independently from one another and assigns the uncertainty metric as the variance of the scores across all models (Seung et al., 1992). Note that empirically, we observed this yields near identical results as the dropout-based approach, so we only present findings for the latter.

#### 3.2.1 Score Calibration

Our model's predicted scores tend to be uncalibrated (as is very typical of neural networks (Guo et al., 2017)), particularly in initial rounds and for infrequent fields due to training data scarcity. We calibrate scores in such a way that picking

a candidate with a calibrated score of, say, $0.6$ yields a $60\%$ probability that it has a positive label (Guo et al., 2017). Without loss of generality, we use a modified version of histogram binning (Zadrozny and Elkan, 2001) and IsoRegC (Zadrozny and Elkan, 2002) to accommodate the highly non-uniform distribution of scores.

By calibrating the scores, threshold selection becomes much more intuitive for the score-based uncertainty metric. For example, if we specify a threshold of $0.5$, we expect that to mean we will select candidates for which the model has a $50\%$ chance of classifying correctly *across all fields*.

### 3.3 Sampling Candidates

Once the uncertainty metric is calculated for each candidate in the unlabeled set, the next step is to select a subset of those candidates for human review. The most obvious method is to select the top-$k$ candidates, thereby selecting the candidates for which the model is most uncertain. In practice, this can lead to sub-optimal results when the model finds many examples for which it is uncertain but may in fact be very similar to one another. The most common approach to break out of this trap is to introduce some notion of diversity in the sampling methodology (Gao et al., 2020; Ishii et al., 2002).

**Combining Top-$k$ and Random Sampling.** A common method is to reallocate the $k$ budget in each round so that a portion of that budget goes towards the top candidates by uncertainty (ensuring we get labels for the most uncertain candidates) and the remaining budget goes towards a random sample of candidates from the unlabeled set (ensuring that some amount of diversity is included in each round). We take a simple approach is to select the top-$k'$ candidates by the uncertainty metric, where $k' < k$, and then randomly sample $k - k'$ candidates from the remaining unlabeled dataset.

**Capping Candidates for Each Document and Field.** An important observation we make about the extraction problem is the following: While a given field typically has multiple candidates in every document, there are usually few positives per document compared to the number of negatives. For example, there are usually many dates in an invoice, and typically only one of them is the *date of invoice*. The uncertainty metrics we defined in Section 3.2 do not take into account this relationship between labels. We leverage this intuition to increase sample diversity by capping the number of

| Domain | # Fields | Splits | # Docs | # Candidates |
|---|---|---|---|---|
| Supply Chain | 18 | Initial-50 | 50 | 11.8K |
| | | Initial-100 | 100 | 24.5K |
| | | Initial-250 | 250 | 58.7K |
| | | Test | 5,019 | 1.2M |
| | | Hidden-label | 10,000 | 2.4M |
| Retail Finance | 11 | Initial-100 | 100 | 76.0K |
| | | Test | 849 | 1.2M |
| | | Hidden-label | 4,000 | 5.6M |
| Tax Forms | 24 | Initial-100 | 100 | 13.4K |
| | | Test | 1,498 | 1.0M |
| | | Hidden-label | 7,500 | 5.1M |

Table 1: Statistics of datasets in three domains.

candidates selected from the same document and field. After ordering the candidates by the chosen uncertainty metric, if we were to simply select the top-$k$ candidates, we might end up selecting too many candidates for the same document and field. Instead, we select at most $m$ candidates for each document and field, $m$ being a tunable hyperparameter we can adjust on a per-field basis. This ensures that we spread the annotation budget over more documents and fields.

### 3.4 Automatically Inferring Negatives

After candidates have been selected and labeled, we merge the newly-labeled candidates into our training set. At this point, there is another opportunity to draw additional value from the unlabeled corpus by utilizing the structure of the extraction problem, in particular, for fields that are defined in the domain's schema to only have a single value per document (such as a document identifier, statement date, amount due, etc.). The key insight here is that when a positive label is revealed via selective labeling, we can infer negative labels for some remaining candidates in the document. If the schema indicates that a particular field is non-repeating, we can automatically infer that all of that field's remaining candidates in the document are negative.

### 4 Datasets and Setups

To evaluate the performance of our proposed methods, we use datasets belonging to three different domains, summarized in Table 1. The number of fields varies across domains, e.g., the *Tax Forms* dataset has more than twice the fields as the *Retail Finance* dataset. We use hidden-label datasets instead of real unlabeled datasets and simulate the labeling procedure by revealing the labels of the candidates from the hidden-label datasets. We leverage Average E2E Max F1 to evaluate the methods. Further explanations and experimental setups can be found in the Appendix.

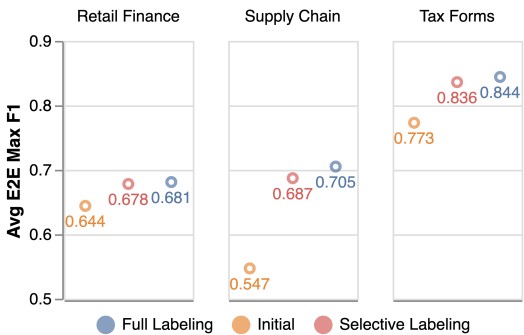

Figure 4: Best performing Selective Labeling as compared to Initial which is trained on just 100 documents and Full Labeling in which the hidden-label dataset (used in Selective Labeling) is fully used in training.

# 5 Results

In this section, we provide evidence to prove that selective labeling reduces the annotation cost by 10X in different domains and analysis to support the design choices including number of selection rounds, selection and sampling strategies.

## 5.1 Best Performance on Different Domains

We train three initial models on a randomly sampled and labeled set of 100 documents for each domain. For example, as shown in Figure 4, the initial model for the *Supply Chain* domain achieves 0.547 F1 on the test dataset. We fine-tune the initial model on a fully labeled 10k document dataset (i.e., the hidden-label set from Table 1, in which for the purposes of this analysis we use its true labels), resulting in an F1 score of 0.705. The performance gap between these two models is thus 0.158.

Starting from the same initial model, we apply our best selective labeling strategy (which we discuss in the following sections) to reveal the labels from a subset of candidates that comprises only 10% of the annotation cost of fully labeling the hidden-label dataset. For the *Supply Chain* domain, this achieves an F1 score of 0.687, which closes the performance gap by 89%. Similarly, we close the gap by 88% and 92% for the *Retail Finance* and *Tax Forms* domains, respectively. This demonstrates that our method can dramatically decrease the annotation cost without sacrificing much performance and can be generalized well to other document types.

## 5.2 Selection Metrics

In Figure 5a we plot per-round performance of two selection metrics in the *Supply Chain* domain given the same set of documents and annotation

budget (i.e, 10% cost) and using the top-$k$ sampling methodology. We observe that not only is computing score distances as the uncertainty indicator much more computationally efficient than variance-based metrics ($10\times$ faster), but it also significantly outperforms the latter as well. As we exhaust the budget over time, the advantage of score distance becomes more obvious.

## 5.3 Sampling Methodology

Figure 5b compares performance across different sampling methodologies. As one might expect, pure random sampling is far worse than any other approach – we believe the initial model is confident in predicting a large quantity of candidates (especially the negatives), and randomly sampling from them does not obtain much useful knowledge.

The top-$k$ strategies produce much more impressive results. Furthermore, we observe in later rounds that injecting some diversity via randomness achieves slightly better performance than the vanilla top-$k$ approach. We believe this mimics the aggregation of exploitation (top-$k$) and exploration (random) processes, proven to be beneficial in reinforcement learning applications (Ishii et al., 2002). This also confirms our suspicion that top-$k$ alone can lead us into selecting many uncertain examples which are in fact very similar to one another.

## 5.4 Multi-round Setting

In Figure 5c, we compare 5 learning curves, each of which denotes selecting the same number of candidates in total (10% annotation cost) over a different number of rounds. For example, the 16-round experiment selects $\frac{1}{16}$ of the total budget in each round, while the 1-round experiment utilizes the entire budget in a single round.

As we increase the total number of rounds, the model tends to yield better extraction performance until it peaks at about 12 rounds. This finer-grained strategy usually performs better than coarser ones but the gains become marginal at a higher number of rounds. Interestingly, we find that using up just half the budget in the first 8 rounds of a 16-round experiment achieves slightly better performance than exhausting the entire budget in the 1-round experiment. This comparison underscores the importance of employing a multi-round approach.

## 5.5 Ablation Study

Table 2 presents an ablation study to understand the impact of different diversity strategies. SL rep-

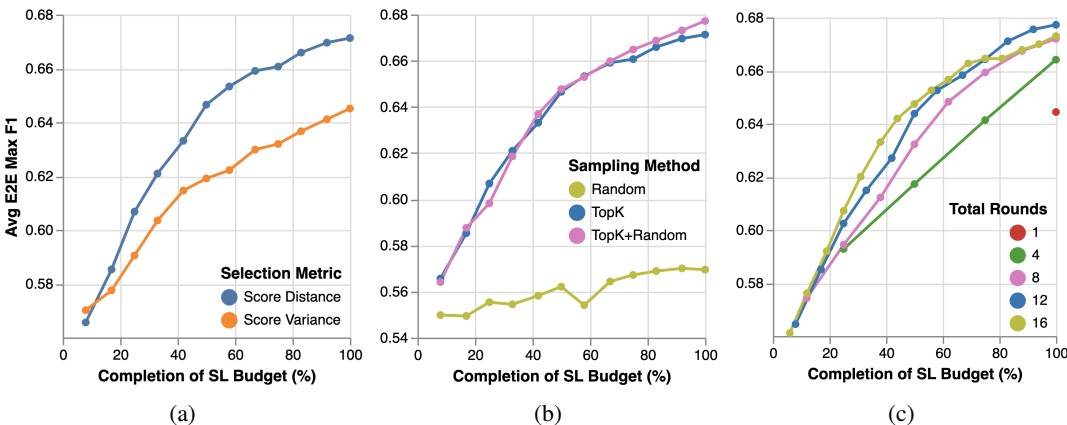

Figure 5: Performance comparisons between (a) selection metrics, (b) sampling approaches, and (c) the rate at which we exhaust the budget through different number of rounds of selective labeling. The x-axis denotes the percentage of the total selective labeling budget consumed. The results are about the *Supply Chain* dataset.

| Models | Avg E2E Max F1 (std.) | Δ |
|---|---|---|
| SL | 0.671 (0.006) | - |
| SL+CS | 0.679 (0.005) | +1.2% |
| SL+CC | 0.675 (0.005) | +0.6% |
| SL+AIN | 0.683 (0.009) | +1.8% |
| SL+CS+CC+AIN | 0.687 (0.005) | +2.1% |

Table 2: Ablation Study. SL denotes selective labeling utilizing the top-$k$ sampling and score distance metric. CS, CC, and AIN represent calibrating scores, capping candidates and automatically inferring negatives.

resents a 12-round selective labeling method using top-$k$ sampling on the score distance metric. We separately add one feature at a time to test the effectiveness of calibrating scores (CS), automatically inferring negatives (AIN) and capping candidates (CC). Results show that every feature improves the model, but we achieve the largest improvement when applying all features in SL+CS+CC+AIN. It is reasonable to conclude that increasing diversity intelligently helps us select more useful candidates than relying on the uncertainty metric alone.

## 5.6 Initial Labeled Dataset Size

Given the dependence of the selective labeling method on an initially labeled small dataset, it is imperative that we evaluate how the approach is affected by the number of documents in this initial dataset. We experiment with initial datasets of 50, 100, and 250 documents in the *Supply Chain* domain using our best selective labeling strategy and a budget equivalent of 10% cost of annotating the "unlabeled" dataset.

Figure 6 indicates that the size of the initial dataset greatly impacts the performance of the model trained solely on those initial training sets, but has starkly less of an impact once we apply selective labeling. We close the performance gap

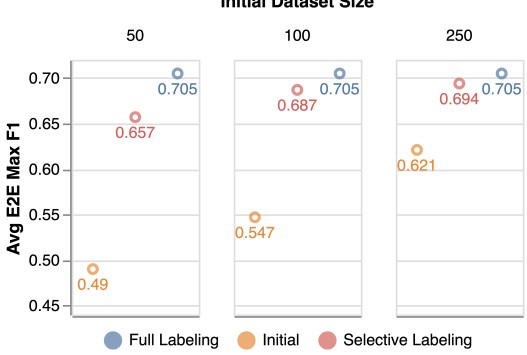

Figure 6: Comparison among three initial dataset sizes in the *Supply Chain* domain. We present the same three approaches as in Figure 4: Initial is trained on the initial dataset alone, Selective Labeling selects the equivalent of 10% annotation cost in candidates, and Full Labeling fine-tunes from the initial model on the full hidden-label data.

by 77%, 89%, and 87%, for initial dataset sizes of 50, 100, and 250, respectively. We can conclude that selective labeling is capable of finding useful candidates to significantly improve the model performance even at a cost of only 10% of the annotation budget. And it is not surprising that the selective labeling gains may suffer when the initial dataset is too small (e.g. 50).

Note that the model can extrapolate to fields that are not present in the initial set of documents. For each document type, a schema is defined to include all types of fields that users may be interested in. The model can generate candidates no matter if the field exists in the initial document set or not, as long as the field is included in the schema.

## 5.7 Per-field Extraction Analysis

We examine the extraction performances of eight fields from the *Supplier Chain* document type in

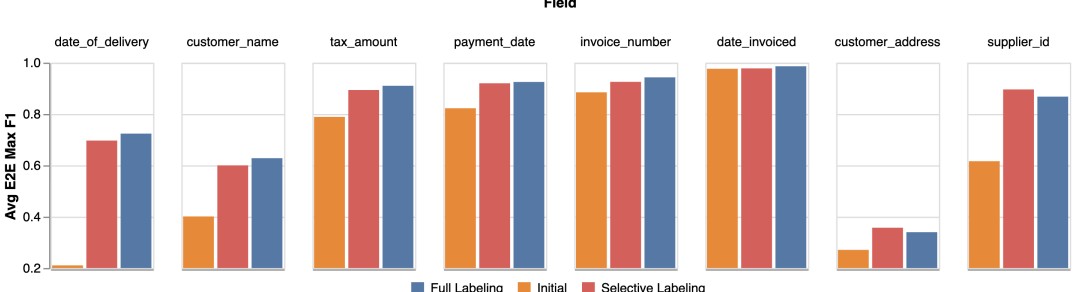

Figure 7: Per-field comparison among Initial, Selective Labeling and Full Labeling. Initial is trained on the initial dataset alone, Selective Labeling selects the equivalent of 10% annotation cost in candidates, and Full Labeling fine-tunes from the initial model on the full hidden-label data.

Figure 7 (initial dataset size is 100) to better understand where selective labeling works well. The recall of candidate generation for these fields varies from 30% to 99% showing that selective labeling works even when candidate generation is not perfect. We observe that the big gap between Initial and Full Labeling is almost completely closed by selective labeling in fields such as *date_of_delivery* and *customer_name*. Unsurprisingly, the algorithm results in strong improvements for fields where the initial model hasn't seen enough examples. For frequent fields such as *date_invoiced* and *invoice_number*, the initial model performs well, and there is not much room for improvement. Consequently, few candidates are selected and the resulting Selective Labeling model remains competitive on these fields.

## 6 Related Work

**Form Extraction.** There have been numerous recent studies on information extraction for form-like documents. Existing approaches either individually categorize every text span in the document (Majumder et al., 2020; Zhang et al., 2021) or formulate the task into a sequence modeling problem (Aggarwal et al., 2020; Lee et al., 2022; Garncarek et al., 2021; Xu et al., 2021) and encode texts, layouts, and visual patterns into feature space. While these approaches produce state-of-the-art extraction systems, they require large amounts of labeled training data to do so. We do not propose a new model architecture but instead, focus on the cost of acquiring labeled data for such extraction models.

**Active Learning.** We refer to (Settles, 2009; Fu et al., 2013; Ren et al., 2021) for an extensive review of the literature. Two popular approaches for requesting annotation are uncertainty-based selection (Ko et al., 1995; Culotta and McCallum, 2005) and committee-based selection (Gal and Ghahra-

mani, 2016; Kirsch et al., 2019; Bengar et al., 2021). Researchers seek to increase the diversity by forcing the selection to cover a more representative set of examples (Yang et al., 2017; Yin et al., 2017; Sener and Savarese, 2018) or incorporating discriminative learning (Gissin and Shalev-Shwartz, 2019). Researchers have studied combining active learning with deep learning. Most of the advanced strategies such as Coreset (Sener and Savarese, 2018), Dropout (Gal and Ghahramani, 2016), Discriminative Active Learning (Gissin and Shalev-Shwartz, 2019). (Zhang et al., 2017; Prabhu et al., 2019; Siddhant and Lipton, 2018; Zhang and Plank, 2021; Yuan et al., 2020; Ash et al., 2020; Yin et al., 2017; Shui et al., 2020) studies deep active learning on various NLP tasks. While we don't focus on inventing new active learning methods, we are the first to customize existing methods to reduce annotation costs in form-like document extraction.

## 7 Conclusion and Future Work

We propose *selective labeling* that *dramatically cuts down the primary dataset annotation bottleneck* in developing extraction models for visually rich documents. There are several future avenues for investigation. First, we simplified the annotation task to a binary "yes/no" question. Another approach is to allow the annotator to either accept the candidate annotation, or *correct* it – either by deleting it or by adjusting the bounding box. For certain text fields it can be valuable to adjust spans to include/exclude details like salutations from a name field ("Mr.", "Dr." etc.) or names from an address. The cost model for such an option is more complex than "yes/no", but can be used to build on the results in this paper. Second, many recent approaches (Xu et al., 2021; Lee et al., 2022) treat this as a sequence-labeling problem and use a layout-aware language model. Adapting selec-

tive labeling to a sequence-labeling model requires tackling several problems: a) getting uncertainty estimates for a given span from a sequence labeling model, b) training a sequence labeling model using partially labeled documents, and c) optionally, eschewing candidate-generators entirely and generating both candidate-spans and their uncertainty estimates form the sequence labeling model. We hope to explore the multiple ways to tackle each of these problems in future work.

## 8   Limitations

Within the scope of this paper, the proposed method is limited to utilizing combinations of candidate generators and scorers. As explained in Section 7, many recent attractive approaches treat document extraction as a sequence labeling problem using a layout-aware language model. This model family is attractive because it does not require a candidate generator. However, constructing selective labeling on sequence labeling models is not a simple task, as we must figure out how to obtain an uncertainty estimate for each span from a sequence labeling model, how to define spans without a candidate generator, and how to train the model with partially labeled documents, etc.

We understand the limitation of the availability of datasets. We are currently unable to open-source them since the datasets contain proprietary information (such as vendors and suppliers) that prevent us from sharing publicly. We use internal datasets in this work because they reflect the real-world needs of our institution and its customers better than public datasets. Compared to the few available public datasets, such as FUNSD (Jaume et al., 2019) and CORD (Park et al., 2019), our internal datasets are reflective of real-world data set sizes, which are appropriate for model training and selective labeling, and have more realistic document complexity (considering the rich schema, layout-rich documents, diverse templates, high-quality OCR results and token level annotation). Additionally, we list all important information needed to reproduce the methods in the Appendix, including the annotation strategies, model dependencies, and selected hyperparameters.

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

# A   Appendix

## A.1   Experimental Setups

To explore how the size of the initial labeled dataset impacts our methods, we create three initial splits for the *Supply Chain* domain with 50, 100, and 250 documents. In all of our experiments, we split the train set into 80-20 training-validation sets. The validation set is used to pick the best model by AUC-ROC, and we use the test split to report the performance metrics. We train using the Rectified Adam (Liu et al., 2020) optimizer and measure AUC-ROC on the validation set to decide whether to trigger early stopping after 3 epochs of no improvement. The binary classifier has 330k parameters and each set of experiments trained within 4 hours on a NVIDIA Tesla P100 GPU. We apply grid search to tune the hyperparameters. The most performant hyperparameter values are listed in Table 3.

## A.2   Evaluation Metrics

We evaluate our methods by measuring the overall extraction system's performance on the test set using the maximum F1 averaged across all fields, denoted as "Average E2E Max F1" in (Majumder et al., 2020). Here we define the "Average E2E Max F1" metric in sufficient detail. "F1" is the harmonic mean of precision and recall. By varying the threshold at which we operate the binary field classifier, we vary precision and recall, and thus the F1. "Maximum F1" simply refers to the highest F1 that we can achieve with a trained field classifier – put another way, this is the point along the precision-recall curve at which the F1 score is highest. We take the "maximum F1" of each field in the schema and average them to compute the "Average E2E Max F1", thereby providing a single metric by which we can evaluate overall performance. We optimize for macro F1 which weighs each field equally irrespective of frequency since macro F1 is the typical goal for most extraction applications. Every reported F1 score is further averaged over 10 independent runs to account for variability. All F1 scores are generated by comparing the extractions with the ground truth. If a field has a poor candidate generator, its final recall can obviously not exceed the recall of the candidate generator.

## A.3   Annotation Budgets

We acquired stats from our team of annotators on how long the classic annotation takes for various

| Hyperparameter | Range explored | Best performer |
|---|---|---|
| learning rate | 0.0001-0.1 | 0.001 |
| dropout rate | 0.1-0.5 | 0.1 |
| batch size | {64, 128, 256} | 128 |
| top-$k'$ uncertain candidates | 0.7-1.0$k$ | 0.9$k$ |
| $m$ candidates each field doc | 1-3 | 1 |

Table 3: Hyperparameter selection.

document types. We found it averaged 6-8 min for an annotator to label a single-page document with fewer than 20 fields while it averaged 10-30 min for an annotator to label a multi-page document with 25 fields. So we picked a very conservative value (6 min) as the estimated time of labeling one document in this paper. We employ two annotation methods: the classic annotation method, which is always applied to the initial training set, and the proposed "yes/no" method, which is applied during the selective labeling procedure on the unlabeled dataset. The annotation budget is computed based on the time needed to annotate a full document and to answer a yes/no question. Targeting 10% of the cost to fully label the unlabeled dataset via the classic annotation method, translates to selectively labeling 36k, 14k, and 27k "yes/no" questions for *Supply Chain*, *Retail Finance*, and *Tax Forms* domains according to the estimation of same amount of annotation hours. If we bootstrap the model using the classic annotation workflow on a small number of documents, we simply subtract that cost from the budget for selective annotation.

## A.4   Imperfect Candidate Generation

We believe that the problem of imperfect candidate generation requires more discussion. We build our Selective Labeling framework on the model architecture introduced in (Majumder et al., 2020; Tata et al., 2021) where they have already demonstrated that high-recall candidate generators can be built (and are in use) for many fields like *numbers, prices, dates, names of people, places organizations* (using canned named-entity annotators), *addresses* (using canned address detectors), *alphanumeric strings* (canned regexes), etc. We denote the recall of candidate generation as coverage (to distinguish this from recall of extractions) and present this for a few fields in Table 4. For simple fields, the coverage is indeed in the 90s (*date_of_delivery, purchase_order*), but for other fields it is lower. Having a candidate generator with low coverage indeed limits the recall of the final extractions and therefore the final F1 score for that field. A key

| Field | Coverage | F1 w/o SL | F1 w/ SL | Δ |
|---|---|---|---|---|
| *date_of_delivery* | 0.932 | 0.690 | 0.759 | 10.00% |
| *purchase_order* | 0.992 | 0.884 | 0.963 | 8.94% |
| *customer_address* | 0.484 | 0.293 | 0.363 | 23.89% |
| *customer_name* | 0.715 | 0.526 | 0.621 | 18.06% |

Table 4: The F1 performance comparison between w/o Selective Labeling and w/ Selective Labeling on four fields. (Coverage: Candidate Generator Recall)

| | Cost | Supplier Chain | | Retail Finance | | Tax Forms | |
|---|---|---|---|---|---|---|---|
| | | F1 | gap closed | F1 | gap closed | F1 | gap closed |
| SL | 0% | 0.547 | 0.0% | 0.644 | 0.0% | 0.773 | 0.0% |
| | 10% | 0.687 | 88.6% | 0.678 | 91.9% | 0.836 | 88.7% |
| | 20% | 0.704 | 99.4% | 0.682 | 102.7% | 0.844 | 100.0% |
| | 30% | 0.706 | 100.6% | 0.686 | 113.5% | 0.845 | 101.0% |
| FL | 100% | 0.705 | 100.0% | 0.681 | 100.0% | 0.844 | 100.0% |

Table 5: Comparisons of the extraction performance and gap closed by SL when consuming different annotation costs on three datasets. SL 0%, 10%, and FL 100% correspond to the Initial, Selective Labeling, and Full Labeling stats in Figure 4. (SL: Selective Labeling, FL: Full Labeling)

contribution of this paper is that even with imperfect candidate generation, that is, on fields with low candidate coverage, Selective Labeling allows us to deliver big improvements to the final extraction F1 score for that field. See F1 scores before and after selective labeling for fields in the Supply Chain dataset in Table 4.

For simple extraction types such as *purchase_order* (numbers) and *date_of_delivery* (dates), Selective Labeling can improve their extraction performance by 10%, thanks to high candidate generation coverage. For fields with low candidate generation coverage such as *customer_name* and *customer_address*, Selective Labeling is still able to find the uncertain candidates and dramatically improve their extraction F1 by 24% and 18% respectively. Finally, even with perfect candidate generation (100% coverage), naively using candidate-level labeling (yes/no questions) instead of the classic annotation workflow ends up being more expensive. As we explain in Section 2.1.2, this is because we typically see dozens of candidates for many fields, and answering yes/no for each of these candidates ends up taking longer than the classic annotation workflow.

## A.5 Increasing Annotation Budget

We gradually increased the annotation budget and observed the corresponding results in Table 5. Here we compared the extraction performance and gap closed using 10% (same as stats in Figure 4), 20%,

and 30% of the annotation budget. If our goal is to close the gap by about 90%, then 10% of the labeling cost is enough to achieve the goal (for all three data sets). If our goal is to close the gap by 99%, then 20% of the labeling cost is sufficient for three datasets. The performance of selective labeling can even exceed the full labeling setup (see results for 30% of the labeling cost). We believe that's because noisy annotations exist in the full dataset and our Selective Labeling algorithm avoids selecting a few incorrect annotations that can confuse the model especially when the model has already determined predictions with low uncertainty that are contrary to the ground-truth annotation.

## A.6 Candidate Generator

As explained in Section 2.2, we use an extraction system similar to the architecture described in (Majumder et al., 2020). As shown in Figure 2, this architecture consists of two stages: candidate generation and candidate classification. Candidate generators are used to identify potential values for a given field. The generators use a cloud-based service[1] to identify text spans in the OCR text that are instances of the corresponding type. The system leverages an internal library[2] of text annotators that was developed for web-search tasks. These text annotators are used to extract features from text that can be used to identify potential values for a field. In addition to the internal text annotators, the system may also use several open-source entity detection libraries. These libraries are used to detect common types of entities, such as names, dates, currency amounts, numbers, addresses, URLs, etc. Entities are identified by matching text against patterns that are known to represent these entities. For example, any date that is found in an invoice is considered a possible value for any of the date fields in the target schema, such as invoice_date, due_date, and delivery_date.

## A.7 Neural Scoring Model

To predict whether an extraction candidate is a valid value for a given target field, the scorer model takes the target field and the extraction candidate as input and outputs a prediction score. The model is trained and evaluated as a binary classifier, which means that it predicts whether an extraction can-

---
[1] cloud.google.com/natural-language
[2] NLTK: https://www.nltk.org/

didate is valid or invalid. The features of each extraction candidate used in the scorer model are its neighboring words and their relative positions. The model learns a dense representation for each extraction candidate using a simple self-attention based architecture. This representation captures the semantics of the extraction candidate. The model also learns dense representations for each field in the target schema. These representations capture the semantics of the fields. Based on the learned candidate and field representations, each extraction candidate is scored based on the similarity to its corresponding field embedding. The model is trained as a binary classifier using cross-entropy loss. The target labels are obtained by comparing the candidate to the ground truth. Details of the model architecture can be found in (Majumder et al., 2020).