# OpenReview forum: "Selective Labeling: How to Radically Lower Data-Labeling Costs for Document Extraction Models"
_EMNLP/2023/Conference — EMNLP 2023 Main_

### Official Review · Reviewer_5HoU · 2023-08-05

**Soundness:** 3

**Excitement:**

2: Mediocre: This paper makes marginal contributions (vs non-contemporaneous work), so I would rather not see it in the conference.

**Missing References:**

Two step extraction was proposed in the following paper.

- Named Entity Extraction using AdaBoost (CoNLL’02)

 The following paper proposed to handle ambiguous annotation for sequence labelling, which can also be applied to partial annotation.

- Training Conditional Random Fields Using Incomplete Annotations. (COLING 2008)

 A partial annotation-based active learning for sequence labeling was addressed in the following paper.

- Active Learning with Subsequence Sampling Strategy for Sequence Labeling Tasks. (Information and Media Technologies 2011)

**Paper Topic And Main Contributions:**

This paper proposes selective labeling as a solution to visually rich documents like invoices, receipts, bills, tax forms, and so on. The key insight is to simplify the labeling task to provide “yes/no” labels for candidate extractions predicted by a model trained on partially labeled documents. This paper combines the labeling task of "yes/no" with a custom active learning strategy to find the predictions that the model is most uncertain about. Experimental results on document types from different domains that the labeling can reduce the cost of acquiring labeled data with a negligible loss in accuracy.

**Questions For The Authors:**

(A)	The reviewer thinks similar approaches of this paper have been tackled in problems of sequence labeling such as named entity recognition. Are there any specific problems of document extraction not addressed in other sequence labeling problems?

**Reasons To Accept:**

Evaluation of two step annotation consisting of selection of candidate from extracted results and annotating labels of the selected candidates.

**Reasons To Reject:**

In named entity recognition, which is categorized as sequential labeling problem as in document extraction of this paper, similar problems have been studied.However, they are not mentioned. Some of them are listed in the missing reference section. The reviewer feels that the combination of existing methods including the missing reference ones and active learning consist of this paper's method.

**Reproducibility:**

1: Could not reproduce the results here no matter how hard they tried.

**Reviewer Confidence:**

2: Willing to defend my evaluation, but it is fairly likely that I missed some details, didn't understand some central points, or can't be sure about the novelty of the work.

---

> ### Author Rebuttal · Authors · 2023-08-29
>
> We appreciate the reviewer’s thoughts and comments. In this rebuttal, we mainly respond to the two concerns of the reviewer:
>
> *1.  Are there any specific problems of document extraction not addressed in other sequence labeling problems?*
>
> Yes, the problems of document extraction that were not addressed in other sequence labeling tasks are in three folds:
>
> (1) Paper shown in the “missing reference” (Active Learning with Subsequence Sampling Strategy for Sequence Labeling Tasks, Wanvarie et al., 2011) does not tackle the problem of turning the sequence labeling methods into answering Yes/No annotation questions, which is actually the key for annotation cost reduction in our approach. **Answering Yes/No questions, compared to labeling text spans (partial sequence), is much faster and adds less cognitive burden to the annotators**;
>
> (2) We believe that adapting selective labeling to work with sequence labeling models is an important problem, and indeed we're working on it, but that is not the focus of this paper (explained in more detail in the next question). In this paper, we're focused on showing that we can cut the cost of labeling for the two-stage binary classifier based approaches (Majumder et al., 2020) by 10x. We hope to demonstrate the same for sequence labeling approaches in a future paper. We focus on this binary classifier-based approach because it is less likely to underfit on small annotated datasets than prevailing sequence labeling models such as FormNet (Lee et al., 2022) or LayoutLM ( (Xu et al., 2021) and annotating each candidate is more natural to be converted to a Yes/No question;
>
> (3) Active learning methods designed in the “missing references” (such as Wanvarie et al., 2011) cannot fit the recently developed layout-aware sequence labeling methods, e.g. how to encode surrounding layout information from the visually complex document to select partial text spans for annotation remains an open question.
>
> $\newline$
>
> *2. Why didn't the author mention the sequence labeling related papers (as recommended in the missing reference section)?*
>
> First of all, we did not mention the recommended references because the proposed annotation cost reduction strategy (selective labeling) is not built on top of any sequence labeling methods, but instead relying on a widely used binary classifier-based solution (Majumder et al., 2020) for visually complex document extraction. This extraction solution finds candidates of each field type for each document, and then applies a binary classifier to predict whether the candidate belongs to the type or not. This method has a relatively small number of parameters, which makes it easy to train on a limited amount of data but can still achieve good extraction performance, so it has been widely applied in the industry. **Most importantly, the two-stage binary classifier-based method can be naturally converted to a yes/no annotation task of low cognitive burden. This is the foundation of selective labeling.** There is an urgent need to explore efficient annotation strategies for the binary classifier-based methods, which motivates this paper.
>
> Though the sequence labeling method is also a prevailing technique for document extraction (it converts the document into a long sequence and applies a named entity recognition strategy to tag each token of the sequence), we did not explore how to reduce the cost for any sequence labeling methods in this paper. And it is non-trivial to adapt the annotation strategy designed for one type of document extraction method to another. For example, to enable the proposed selective labeling on a sequence labeling method, we need to further explore how to pick text spans from a long sequence and how to estimate their uncertainty. We also need to find ways to apply Yes/No annotation tasks to the selected partial sequences. As we discussed in Section 7 of this paper, we agree that reducing annotation cost for the state of the art layout-aware sequence labeling methods (e.g. Lee et al., 2022) is an interesting topic to explore in the future. We are grateful to the reviewer for providing us with some references, which can really be a good start of this future work. Actually, we have already made some preliminary progress in this direction, and we found that selective labeling on sequence labeling models can achieve similar cost reduction effects as what we observed in our paper. We would be happy to share more experimental results if the reviewer is interested in this thread.

---

### Official Review · Reviewer_b6MD · 2023-08-05

**Soundness:** 4

**Excitement:**

3: Ambivalent: It has merits (e.g., it reports state-of-the-art results, the idea is nice), but there are key weaknesses (e.g., it describes incremental work), and it can significantly benefit from another round of revision. However, I won't object to accepting it if my co-reviewers champion it.

**Paper Topic And Main Contributions:**

In this paper, the authors propose a method called “selective labelling” to more effetely produce human annotated labels for training models for visually extracting information from documents, like invoices, receipts, bills tax forms, etc. The main idea is to leverage the active learning paradigm and with the help of the training model simplify the complex labeling problem. Instead of directly asking humans to annotate fields in the documents, the training model first selects extraction candidates, and humans simply agree or disagree with the annotations, thereby reducing the mental burden of the annotation task. To validate their approach, the authors conducted experiments on three types of documents (supply chain, retail finance, and tax forms) and concluded that their method could reduce the cost of acquiring labeled data by up to 10 times while maintaining similar performance.

Overall, I find the paper well-written, well-organized and well-motivated. I believe the main contribution in the paper is a simple but effective methodology for cheaply acquiring labeling data for training models in visual extraction tasks.


**Questions For The Authors:**

- **A:** Can the model extrapolate to fields that are not present in the initial set of documents?
- **B:** Could you provide clarification on how the score calibration is performed?


**Reasons To Accept:**

- **Simple and sound method:** The proposed method is straight forwarded and easily appliable.

- **Extensive evaluation:** The authors deeply investigated the behaviour of the proposed method under multiple circumstances, which revealed insightful results.

**Reasons To Reject:**

- **Felt there are details missing:** Although, the focus is on the SL method itself and not on the extractive models, it would be beneficial if the paper becomes more self-contained by briefly addressing the candidate generator and scorer model used during the experiments, instead of providing only a brief glimpse. Additionally, exploring the synergy of different models and the proposed SL method could add an interesting perspective to the paper.

- **Missing limitation:** After reading the paper, I felt that there was one limitation that was not addressed. From what I understood, if a rare field is not present in the initial set of documents, then it would be impossible for the extraction model to generate candidates for that missing field, meaning that field would never receive an annotation.

- **Hard reproducibility and no codebase:** The paper's reproducibility is challenging due to the private nature of the chosen datasets (as correctly mentioned in the limitation section). To overcome this, conducting additional experiments on public datasets would enhance the broader dissemination of the work. Moreover, even though the datasets are private, releasing the codebase could facilitate a larger adoption of the method and encourage benchmarking on other datasets.

**Reproducibility:**

1: Could not reproduce the results here no matter how hard they tried.

**Reviewer Confidence:**

2: Willing to defend my evaluation, but it is fairly likely that I missed some details, didn't understand some central points, or can't be sure about the novelty of the work.

**Typos Grammar Style And Presentation Improvements:**

Consider adding an algorithmic description of the SL steps.

Consider adding in the legend of Fig 5 that the results are about the supply chain dataset.

---

> ### Author Rebuttal · Authors · 2023-08-28
>
> Thank you for taking the time to review our paper. We appreciate your thoughtful comments and suggestions, which have helped us to improve the quality of our work.
>
> *Q1. Felt there are details missing about the candidate generator, scorer model and the exploration of SL with different models.*
>
> We appreciate the suggestion that the paper should become more self-contained by briefly addressing the candidate generator and scorer model. We will add more details of both candidate generator and scorer model into the paper.
>
>
>
> *Q2: Can the model extrapolate to fields that are not present in the initial set of documents? (Missing limitation: if a rare field is not present in the initial set of documents, then it would be impossible for the extraction model to generate candidates for that missing field.)*
>
> Yes, the model can extrapolate to fields that are not present in the initial set of documents. For each document type, a schema is defined to include all types of fields that users may be interested in. **The model can generate candidates no matter if the field exists in the initial document set or not, as long as the field is included in the schema.** If it’s not in the initial document set, there’s actually a higher probability that more candidates of this field will be selected for annotation. So this is not a limitation and we will clarify it in the revision paper.
>
>
> *Q3: Hard reproducibility and no codebase.*
>
> Thank you for your feedback on open-sourcing the codebase for our SL framework. We understand that this would be helpful for the reproduction of the framework, and we are committed to making our research as reproducible as possible. We will try to get approval to open-source the SL-related codes, including how we measure the uncertainty of candidates, and how we calibrate the prediction scores, etc. We have also provided detailed experimental settings in the paper, including hyperparameter choices, model training details, and evaluation metrics, to enhance the reproducibility of our results. In the meantime, the reproduction details for the candidate generator and scorer can be found in its original paper (Majumder et al., 2020).
>
>
> *Q4: Could you provide clarification on how the score calibration is performed?*
>
> We use a modified version of histogram binning and IsoRegC to accommodate the highly non-uniform distribution of scores. We compute calibration curves using the labeled training dataset by bucketing the candidates based on score. Based on our knowledge of the score distribution, we made two design choices. (1) The vast majority ($>90\%$) of our candidates are negative and most of them have very low scores ($<10^{-3}$), while the region of interest to us when calibrating the scores is the rest ($[10^{-3}, 1]$). In calculating bin edges, we exclude all candidates with scores that are smaller than a threshold ($10^{-3}$). All the scores below this threshold are placed in the first bin ($[0, 10^{-3})$). Since the vast majority of candidates get excluded by this filter, the remaining bins have a much higher resolution. (2) We use equal-frequency bins rather than equal-width bins because of the highly non-uniform distribution of scores, even within the score region of interest -- in other words, each bin has roughly the same number of scores, except the first bin. Once binned, calibration curves are computed for each field by interpolating between the curves prevalence (i.e., the proportion of candidates in each score bin that are positive) and the median scores for all the score bins. By calibrating the scores, threshold selection becomes much more intuitive for the score-based uncertainty metric.
>
>
> *Q5: Consider adding an algorithmic description of the SL steps. Consider adding in the legend of Fig 5 that the results are about the supply chain dataset.*
>
> Thank you for these suggestions. We added the descriptions in the revised paper.

---

### Official Review · Reviewer_Gbq2 · 2023-08-05

**Soundness:** 4

**Excitement:**

4: Strong: This paper deepens the understanding of some phenomenon or lowers the barriers to an existing research direction.

**Paper Topic And Main Contributions:**

The paper presents two significant contributions to reducing data-labelling costs for document extraction models. The first contribution involves simplifying the human annotation task to a binary problem. Secondly, it introduces customised active learning at the document-field granularity level.

**Questions For The Authors:**

Have you considered releasing the codebase for your approach?

**Reasons To Accept:**

This paper focuses on a practical problem concerning information extraction from visually rich documents while addressing cost issues with training and human annotations. Although it does not provide a SOTA model, it provides a novel combination to address the stated research problem.

**Reasons To Reject:**

While the authors acknowledge its limitations on dataset release, it would benefit from additional details regarding dataset creation and comparison with other baseline datasets. Furthermore, a comparison with other state-of-the-art models would strengthen the paper's credibility and impact.

**Reproducibility:**

2: Would be hard pressed to reproduce the results. The contribution depends on data that are simply not available outside the author's institution or consortium; not enough details are provided.

**Reviewer Confidence:**

4: Quite sure. I tried to check the important points carefully. It's unlikely, though conceivable, that I missed something that should affect my ratings.

---

> ### Author Rebuttal · Authors · 2023-08-28
>
> We are grateful for the reviewers' time and effort in providing such detailed feedback.
>
> *Q1: While the authors acknowledge its limitations on dataset release, it would benefit from additional details regarding dataset creation and comparison with other baseline datasets. Have you considered releasing the codebase for your approach?*
>
> We are grateful for your recommendation to open-source the codebase for our SL framework. We concur that this would be advantageous for the framework's reproducibility, and we are dedicated to making our research as reproducible as possible. We will try to obtain authorization to open-source the SL-related codes, which include how we quantify the uncertainty of candidates and how we calibrate the prediction scores. We will also include adequate experimental settings in the paper, such as hyperparameter choices, model training details, and evaluation metrics, to enhance the reproducibility of our findings.
>
>
> *Q2: Furthermore, a comparison with other state-of-the-art models would strengthen the paper's credibility and impact.*
>
> The state-of-the-art models for document extraction tasks are FormNet (Lee et al., 2022) and LayoutLMv3 [1]. However, these models are transformer-based sequence labeling models, which are outside the scope of this paper. Our goal is not to build a new state-of-the-art document extraction solution, but rather to introduce a novel method for reducing the cost of annotating data for an existing document extraction solution. We propose to use a binary classifier-based scorer model (Majumder et al., 2020) to implement the selective labeling strategy. This is because (a) this smaller model can be trained more quickly and perform even better than large SOTA sequence labeling methods in low-data regimes; and (b) the candidate generation and scoring architecture enables a more efficient and fine-grained annotation style, in which annotators simply answer Yes/No questions to annotate each candidate. As we discussed in the conclusion section, it is non-trivial to adapt the selective labeling strategy to the SOTA sequence labeling methods. Therefore, it is not feasible to compare our method to these methods at this time. However, we believe that applying selective labeling to those sequence labeling methods is an interesting direction for future work.
>
> [1] Huang, Yupan, et al. "Layoutlmv3: Pre-training for document ai with unified text and image masking." Proceedings of the 30th ACM International Conference on Multimedia. 2022.

---

### Meta-Review · Area_Chair_1yG2 · 2023-09-05

**Recommendation:** 5
**Best Paper Recommendation:** No

**Metareview:**

The paper proposes a method for reducing the cost of labeling data for document-level extraction tasks via active learning by training a model to propose candidate regions and having annotators perform a binary annotation task where they agree or disagree with the proposed extractions. R1 and R2 agree that the method is simple and sound, addresses an important research problem, and is sufficiently backed up by appropriate experimentation. The noted limitations of the work include: the datasets used are private, so reproducibility is an issue and the paper could benefit from experimentation on a public dataset; some important details (namely about the candidate selection model) could be added. The main concern from reviewer 3 is that potentially similar approaches have been proposed on other tasks e.g. named entity recognition. The authors provide a detailed response clarifying many of the reviewers concerns about limitations related to reproducibility, generalizability, and model details. In particular, the though the spirit of the method is similar to two-step sequence annotation, the problem under study is starkly different as such tasks (e.g. NER) still require observing partial sequences, while in visually complex document extraction the authors are able to reduce the problem to individual binary decisions. Overall the paper appears to be sound and of interest to those working on this topic.

**Meta-Review:**

The paper proposes a method for reducing the cost of labeling data for document-level extraction tasks via active learning by training a model to propose candidate regions and having annotators perform a binary annotation task where they agree or disagree with the proposed extractions. R1 and R2 agree that the method is simple and sound, addresses an important research problem, and is sufficiently backed up by appropriate experimentation. The noted limitations of the work include: the datasets used are private, so reproducibility is an issue and the paper could benefit from experimentation on a public dataset; some important details (namely about the candidate selection model) could be added. The main concern from reviewer 3 is that similar approaches have been proposed on other tasks e.g. named entity recognition. The authors provide a detailed response clarifying many of the reviewers concerns about limitations related to reproducibility, generalizability, and model details. Overall the paper appears to be sound and of interest to those working on this topic.

---

### Decision · Program_Chairs · 2023-10-07

**Decision:**

Accept-Main

**Comment:**

The paper proposes a method for reducing the cost of labeling data for document-level extraction tasks via active learning by training a model to propose candidate regions and having annotators perform a binary annotation task where they agree or disagree with the proposed extractions. R1 and R2 agree that the method is simple and sound, addresses an important research problem, and is sufficiently backed up by appropriate experimentation. The noted limitations of the work include: the datasets used are private, so reproducibility is an issue and the paper could benefit from experimentation on a public dataset; some important details (namely about the candidate selection model) could be added. The main concern from reviewer 3 is that potentially similar approaches have been proposed on other tasks e.g. named entity recognition. The authors provide a detailed response clarifying many of the reviewers concerns about limitations related to reproducibility, generalizability, and model details. In particular, the though the spirit of the method is similar to two-step sequence annotation, the problem under study is starkly different as such tasks (e.g. NER) still require observing partial sequences, while in visually complex document extraction the authors are able to reduce the problem to individual binary decisions. Overall the paper appears to be sound and of interest to those working on this topic.